# PERSONALIZED LANGUAGE GENERATION VIA BAYESIAN METRIC AUGMENTED RETRIEVAL

## ABSTRACT

Our paper presents a Bayesian adaptation of Retrieval Augmented Generation (RAG) designed to capture the characteristics of each user, encompassing factors such as their educational background and professions. We model each individual's characteristics using specific perturbations of the local metric of the embedding space. This perturbation introduces a crucial shift in the distance evaluation between the query's and the document's embedding, leading to different pertinent rankings of the retrieved documents. We propose a Bayesian learning procedure that assimilates user feedback and continuously enhances our estimation of the user-specific metric. In the beginning, when there is no information about the user, we use a diverse retrieval method for generation. After this burn-in phase, we learn a Bayesian posterior estimate of the metric, and inject this metric into the nearest neighbor search for document retrieval. This additional layer of metric information acquisition leads to empirical improvement in the retrieval quality and in the performance of the generated text on multiple concept explanation tasks.

## 1 INTRODUCTION

In recent years, retrieval-augmented generation (RAG) has emerged as a pivotal approach in natural language processing. This paradigm shift has been catalyzed by the need to address issues, notably the *hallucination* phenomenon frequently associated with Large Language Models (LLMs). Hallucination occurs when LLMs generate information that lacks accuracy or contextual relevance. In response to this challenge, a burgeoning research trend has deviated from the conventional statistical language modeling paradigm. Numerous existing works (Lewis et al., 2020; Karpukhin et al., 2020) have explored innovative methods to enhance LLMs by equipping them with additional capabilities for information retrieval from external documents. This augmentation extends the contextual scope and enhances relevance, effectively mitigating the issue of hallucination.

Thanks to impressive results on zero-shot and few-shot settings of multiple Nature Language Generation tasks, RAG has been used to build many applications, e.g., chatbot (Shuster et al., 2021), open-domain question answering (Lewis et al., 2020), abstractive summarization (Peng et al., 2019), code generation (Hashimoto et al., 2018), among others. More comprehensive surveys on RAG can be found in Li et al. (2022); Mialon et al. (2023); Zhao et al. (2023). However, limited research has explored the ramifications of the cold-start predicament within real-world contexts regarding its effectiveness. For instance, in scenarios where a novel user initiates interaction with a conversational search-engine system to seek explanations on technical concepts, several documents may be relevant but the user may prefer specific writing styles found in certain documents over others. One can easily envision, for example, a chatbot that explains machine learning concepts to undergraduate students. It is often observed that students from different backgrounds (engineering, business, social sciences, etc.) may prefer different levels of technicality to the explanations. The language generation system can leverage this user-specific preference to improve the effective comprehension of intricate concepts to a wider audience. Consequently, this scenario raises a compelling research question: How can user preferences be effectively incorporated into RAG-based conversational systems?

In this paper, our approach diverges significantly from the predominant trajectory observed in the existing literature on RAG and its incorporation with language generation. We aim to construct a comprehensive three-phase solution package for RAG-based conversational systems. The primary

aim of this package is to enable the retrieval of documents that closely align with human preferences, ultimately enhancing the quality of text generation.

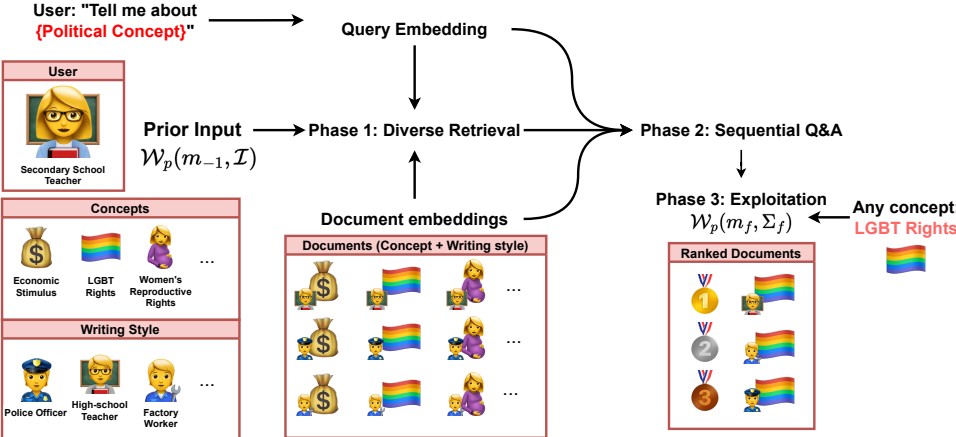

Figure 1: Our solution package consists of three main phases: a determinantal point process sampling for ground-zero preference acquisition (Phase 1), and a sequential question-answering scheme to refine preference belief (Phase 2). Recommendations are tailored to the posterior distribution to capture the preference shifts of the user. Phase 3 is completely exploitative in which the posterior is fixed, and the language generation system aims to match the user's posterior preference.

**Contributions.** We demonstrate that with a limited number of queries, we still can capture the user's preference, and consequentially improve the retrieval and text generation quality. To achieve this goal, we model the user's preference through the metric drift of the embedding space, which governs the ranking of the documents for each input query. More specifically, we postulate that the user's preference for a document's writing style is modeled by a perturbed Mahalanobis metric that deviates from the Euclidean metric. We employ a Bayesian framework to represent our system's belief about the user's metric. Our system sequentially collects feedback using a three-phase scheme:

- At zero-knowledge initialization (Phase 1), we employ a determinantal point process to retrieve a diverse set of summarized and relevant documents to answer the user's queries, and at the same time, collect the user's preferences (pure exploration).

- In Phase 2, we deploy a personalized scheme, consisting of a posterior belief update and a maximal mutual information retrieval system to answer the user's queries (balanced exploration and exploitation).

- In Phase 3, we freeze the posterior distribution and focus on generating answers to the user's queries using the posterior parameters. The posterior indicates a modified metric on the embedding space, which leads to a preference-adapted ranking of the retrieved documents for answering the user's queries.

Our paper focuses on adapting the retrieval mechanism to the the user's preferences, which departs significantly from the conventional generation-based methodologies. Conventional methods include "sparse retriever", denoting a traditional bag-of-words representation encompassing document and query text, e.g., tf-idf, BM25+ (Lv & Zhai, 2011), and "dense retriever", encoding textual information utilizing a neural network (Wang et al., 2020). In our numerical comparison, we will demonstrate that our preference-based retrieval system outperforms conventional methods in both retrieval quality and text generation quality.

**Notations.** The trace of a square matrix $M \in \mathbb{R}^{p \times p}$ is written $\text{Tr}[M] = \sum_{i=1}^{p} M_{ii}$. We use $\mathbb{S}_+^p$ and $\mathbb{S}_{++}^p$ to denote the space of symmetric, $p$-by-$p$ positive *semi*definite and positive definite matrices, respectively. The determinant of a matrix $M \in \mathbb{S}_{++}^p$ is denoted by $\det(M)$.

## 2 BACKGROUND AND PROBLEM FORMULATION

We consider a scenario in which a user submits a series of queries, with each query being transformed into a textual embedding $\{q_t\}_{t \in \mathbb{N}_+}$ within our system. Our database has a collection of $N$ documents, and each individual document is characterized by a corresponding document embedding denoted as $d_i$. Several documents may pertain to identical information; however, they may diverge in their writing style or the analogies employed to elucidate the conveyed information. Our objective is to retrieve the precise document exhibiting the appropriate writing style within the constraints of limited user ratings. Next, we describe how we model the user's preference using a metric on the embedding space, as well as how we are modelling our belief about the user's preference.

### 2.1 USER'S PREFERENCE AND FEEDBACK MODEL

We now describe how we capture the user's preference and how the user interacts with our system. Regarding a query $q_t$, the preference of the user for the $N$ documents, represented by their embeddings $d_i$, is calculated through a distance function $c$ that quantifies the distance between the query $q_t$ and the document $d_i$: the user favors the document if $c(q_t, d_i)$ is less than a certain threshold $\tau$, and the user *dis*favors the document if $c(q_t, d_i)$ is larger than the threshold. Thus, whether the user favors the item or not depends on whether the item is sufficiently close to the user's query. While there is a plethora of choices to parametrize the distance function $c$, we will choose the most simple and intuitive parametrization using the squared Mahalanobis distance in $\mathbb{R}^p$:

$$c_A(q_t, d_i) = (q_t - d_i)^\top A(q_t - d_i),$$

wherein the cost function is fully characterized by a weight matrix $A$ that is symmetric and positive definite. When $A$ is the identity matrix, the Mahalanobis distance becomes the Euclidean distance. Suppose that the user's preference is dictated by a matrix $A_0 \in \mathbb{S}_{++}^p$ and a threshold $\tau$, then for the query $q_t$, the user favors $d_i$ if $c_{A_0}(q_t, d_i) \leq r$, and the user disfavors $d_i$ if $c_{A_0}(q_t, d_i) > r$. We define the feedback function of the user as $R(q_t, d_i, A_0)$ which admits a value $+1$ for a positive preference (favor) and a value $-1$ for a negative preference (disfavor). By a proper normalization of the radius using $A_0 \leftarrow A_0/r$, we have

$$R(q_t, d_i, A_0) = \begin{cases} +1 & \text{if } c_{A_0}(q_t, d_i) \leq 1, \\ -1 & \text{if } c_{A_0}(q_t, d_i) > 1. \end{cases} \tag{1}$$

Thus, without any loss of generality, we suppose that the user's preference and corresponding feedback to our system is completely characterized by a matrix $A_0 \in \mathbb{S}_{++}^p$, and we can now omit the radius $r$. The Mahalanobis preference construction is flexible enough to model the preference variety of the user population. Note that if the weight matrix is $\kappa I$, a positive scaler of the identity matrix, then the user preference is dictated by the Euclidean norm on $\mathbb{R}^p$. This Euclidean norm preference can also be regarded as the *nominal* preference: a Euclidean nearest document will satisfy the user's preference best. Note that the Euclidean nearest neighbor retriever is one of the most popular and fundamental methods in information retrieval. The variation of the user's preference can now be described by shifting the weight matrix $A_0$ away from the scaled-identity matrix.

### 2.2 PROBABILISTIC MODELING OF THE BELIEF ABOUT THE USER'S PREFERENCE

The matrix $A_0$ that determines the user's preference remains elusive to our retrieval system throughout the process. We use a Bayesian framework to represent the system's belief about $A_0$. The Bayesian setting is particularly useful thanks to its power to update the belief as information, or feedback, from the user arrives. Because the true (but unknown) matrix $A_0$ is a $p$-by-$p$ positive semidefinite matrix, we employ the Wishart distribution to characterize our belief.

**Definition 1** (Wishart distribution). *A Wishart distribution on the space of $p$-by-$p$ symmetric positive semidefinite matrices $\mathbb{S}_+^p$ is parametrized by an integer degree of freedom $m \geq p$ and a scale matrix $\Sigma \in \mathbb{S}_{++}^p$. Its density function (with respect to the Lebesgue measure) is*

$$f(A) = \frac{1}{2^{\frac{mp}{2}} \det(\Sigma)^{\frac{m}{2}} \Gamma_p(\frac{m}{2})} \det(A)^{\frac{m-p-1}{2}} \exp(-\frac{1}{2} \operatorname{Tr}[\Sigma^{-1}A]),$$

where $\Gamma_p$ is the multivariate Gamma function

$$\Gamma_p\left(\frac{n}{2}\right) = \pi^{p(p-1)/4} \prod_{j=1}^{p} \Gamma\left(\frac{n}{2} - \frac{j-1}{2}\right).$$

Throughout this paper, we write $\mathbb{P} \sim \mathcal{W}_p(m, \Sigma)$ to denote that $\mathbb{P}$ is a Wishart probability distribution with appropriate parameters. As we have no prior information about the user's preference for the documents, it is reasonable to initialize our belief as a Wishart distribution with identity covariance matrix: $\mathbb{P}_{-1} \sim \mathcal{W}_p(m_{-1}, I)$, with $m_{-1} \geq p$ is an integer number. Choosing an identity matrix as the initial scale matrix of the prior also aligns (up to a positive multiplicative factor) with the nominal preference, as discussed in Section 2.1.

## 3    PHASE 1: DETERMINANTAL POINT PROCESS SAMPLING FOR COLD-START GENERATION

At the beginning of Phase 1, a new user arrives, and we have no prior information about their preferences. To handle this cold-start situation, our system retrieves a set of the most diverse documents related to a specific concept, which are then presented to the user to accumulate user feedback (whether the user favors or disfavors the documents).

Given that documents in this context may be lengthy and complex, we show the user $C_1$ summarized documents, where $C_1$ is a relatively small number, and ask the user for feedback on these summarized documents. At the beginning of this phase, our retrieval system only has access to the query embedding $q_t$ and the initialized documents' embeddings $d_i$, $i = 1, \ldots, N$, but it has no information about the user's preferences on the document's writing style. The goal of Phase 1 is to acquire as much information about the user's preference as possible from a limited number of possible feedback.

To attain this goal, the $C_1$ documents to be retrieved should balance two important criteria:

- (i) relevance: the retrieved document's embedding $d_i$ should be close enough to the query $q_t$,
- (ii) diverse directions: the vectors $d_i - q_t$ should span different angles so that we can learn the shape, or the directions of the eigenvectors, of the weight matrix $A_0$,
- (iii) diverse distances: the distances $\|d_i - q_t\|_2$ should be of different values so that we can learn the magnitude (after normalization to eliminate $r$) of the weight matrix $A_0$.

To meet criterion (i), we first filter a subset of $n$ documents whose embeddings are nearest to $q_t$, for some $n < N$. For criteria (ii) and (iii), we will use a determinantal point process (DPP) to help pick the set of documents that are diverse in both directions and distances. DPP stems from the field of quantum physics: it is used to model the repulsive behavior of Fermion particles Macchi (1975). Subsequently, it has claimed many successes in machine learning tasks Kulesza & Taskar (2012); Affandi et al. (2014); Urschel et al. (2017) and recommendation systems Chen et al. (2018); Wilhelm et al. (2018); Gartrell et al. (2017). We use the following definition of the DPP characterized by the $L$-ensemble.

**Definition 2** ($L$-ensemble DPP). *Given $n$ nearest documents and a positive semidefinite matrix $L \in \mathbb{S}_+^n$, the $L$-ensemble DPP is a distribution over all $2^n$ index subsets $J \subseteq \{1, \ldots, n\}$ such that*

$$\text{Prob}(J) = \det(L_J)/\det(I + L),$$

*where $L_J$ denotes the $|J|$-by-$|J|$ submatrix of $L$ with rows and columns indexed by the set $J$.*

We will construct two matrices to capture each of the above diversity criteria. For diversity of the directions, our system computes the kernel matrix $L^{\text{dir}}$ with the elements

$$L_{ij}^{\text{dir}} = a_i^\top a_j \qquad \forall i, j = 1, \ldots, n, \quad \text{where} \quad a_i = (d_i - q_t)/\|d_i - q_t\|_2 \in \mathbb{R}^p \qquad \forall i = 1, \ldots, n.$$

Each vector $a_i$ is a direction vector of unit length emancipating from the current query embedding $q_t$ towards the embedding of the document $d_i$. The matrix $L^{\text{dir}}$ is a Gram matrix of the linear kernel,

also known as the cosine similarity matrix, and thus $L^{\text{dir}}$ is positive semidefinite. To promote the diversity in the distances, our system computes the kernel matrix $L^{\text{dist}}$ with

$$L_{ij}^{\text{dist}} = \exp(-|\text{dist}_i - \text{dist}_j|^2), \quad \text{where} \quad \text{dist}_i = \frac{\|d_i - q_t\|_2}{\max_j \|v_j - q_t\|_2} \in \mathbb{R}_+ \quad \forall i = 1, \dots, n.$$

Above, $\text{dist}_i$ is the normalized distance from $q_t$ to $d_i$. The matrix $L^{\text{dist}}$ is a Gram matrix with a Gaussian kernel; hence, it is positive semidefinite. Then, we aggregate these two kernel matrices to form the joint ensemble as $L = L^{\text{dir}} + L^{\text{dist}}$, which guarantees that $L$ is also positive semidefinite. Notice that our DPP framework can accommodate a much greater level of flexibility: one can fine-tune the performance by adding a kernel width to construct $L^{\text{dist}}$ and constructing the ensemble $L$ by taking a *weighted* sum. Nevertheless, we take a simplistic approach and omit this generalization to avoid over-parametrization.

To generate a list of $C_1$ documents to show to the user to obtain feedback, we will find the maximum-a-posteriori estimate of the DPP with a cardinality constraint. The result from Kulesza & Taskar (2012) suggests that this set of $K$ items solves

$$\max \left\{ \det(L_J) \; : \; J \subset \{1, \dots, n\}, |J| = C_1 \right\}, \tag{2}$$

where $L_J$ is a submatrix of $L$ restricted to rows and columns indexed by the set $J$. Unfortunately, the above optimization problem is NP-hard Kulesza & Taskar (2012), and our system relies on greedy and local search heuristics to find the optimal index set $J$. Details of the heuristics are relegated to the appendix.

## 4    PHASE 2: SEQUENTIAL FEEDBACK COLLECTION FOR BELIEF REFINEMENT

At the beginning of Phase 2, we have collected certain information about the user's preference from Phase 1. For each query in this phase, the system displays $C_2$ texts generated from $C_2$ documents to balance relevancy and information acquisition. We will provide a posterior update scheme to update our belief in Section 4.1. Moreover, for any query in this phase, we can design a more sophisticated retrieval mechanism using the mutual information criterion, as we will explore in Section 4.2.

### 4.1    POSTERIOR UPDATE

Starting with a generic prior distribution $\mathbb{P}_{t-1} \sim \mathcal{W}_p(m_{t-1}, \Sigma_{t-1})$, we aim to find a posterior to internalize the information that we obtain from the user feedbacks. Suppose that for the current query $q_t$, we have collected the user's preference on the list of documents $\mathcal{I}_t$, and the user returns with a positive experience (favor) on a subset $\mathcal{I}_t^+$ (i.e., $R_{ti} \triangleq R(q_t, d_i, A_0) = +1$ for each $i \in \mathcal{I}_t^+$), with a negative experience (disfavor) on a subset $\mathcal{I}_t^-$ (i.e., $R_{ti} \triangleq R(q_t, d_i, A_0) = -1$ for each $i \in \mathcal{I}_t^-$). The information from liked and disliked items can help us find the posterior belief with two desiderata: (i) the posterior belief $\mathbb{P}_t$ should be close to the prior $\mathbb{P}_{t-1}$ so that we do not forget the previous information, (ii) the posterior should reflect the new information in $\mathcal{I}_t^+$ and $\mathcal{I}_t^-$.

To attain desiderata (i), we track the distortion between the prior and the posterior belief using a Kullback-Leibler (KL) divergence. The KL divergence from a probability measure $\mathbb{P}$ to another probability measure $\mathbb{P}_{t-1}$, with $\mathbb{P}$ being absolutely continuous with respect to $\mathbb{P}_{t-1}$, is defined as $\text{KL}(\mathbb{P} \,\|\, \mathbb{P}_{t-1}) \triangleq \mathbb{E}_{\mathbb{P}}[\log(d\mathbb{P}/d\mathbb{P}_{t-1})]$, where $d\mathbb{P}/d\mathbb{P}_{t-1}$ denotes the Radon-Nikodym derivative. To achieve desiderata (ii), we posit that the posterior should assign a high probability to the events

$$R_{ti}(c_A(q_t, d_i) - 1) \leq 0 \qquad \forall i \in \mathcal{I}_t^+ \cup \mathcal{I}_t^-.$$

Notice that $R_{ti}$ has a value of either $+1$ or $-1$, and the above events align with the preference model postulated in (1). Let us first consider the case of a disfavor feedback when $R_{ti} = -1$: in this case, the posterior should have a small value for $\mathbb{P}(c_A(q_t, d_i) \leq 1)$. In a similar argument, for a favor feedback when $R_{ti} = +1$, the posterior should have a small value for $-\mathbb{P}(c_A(q_t, d_i) \leq 1)$. Consequently, we propose to assign the posterior belief to the optimal solution of the stochastic optimization problem

$$\begin{aligned} \min \quad & \text{KL}(\mathbb{P} \,\|\, \mathbb{P}_{t-1}) - \tau \sum_{i \in \mathcal{I}_t^+ \cup \mathcal{I}_t^-} R_{ti} \log \mathbb{P}(c_A(q_t, d_i) \leq 1) \\ \text{s.t.} \quad & \mathbb{P} \sim \mathcal{W}_p(m, \Sigma), \; m \in \mathbb{N}_+, \; \Sigma \in \mathbb{S}_+^p \\ & p \leq m \leq m_{t-1}. \end{aligned} \tag{3}$$

Problem (3) imposes a parametric Wishart form for the posterior distribution, and it finds the best values of the degrees of freedom $m$ and the scale matrix $\Sigma$ that can minimize the sum of the divergence from the prior and the weighted *log*-probability (or *log*-likelihood) values. The weight $\tau > 0$ indicates the balance between sticking to the prior or adhering to the feedback received for query $q_t$. We explicitly put a bound $m \leq m_{t-1}$ so that the degrees of freedom decrease as time progresses, and the posterior distribution exhibits certain concentration behavior.

We now discuss how we solve problem (3). The KL divergence between two Wishart distributions is known in closed form according to Soch et al. (2020): Let $\psi_p$ denote the multivariate digamma function, the KL divergence from $\mathbb{P}_2 \sim \mathcal{W}_p(m_2, \Sigma_2)$ to $\mathbb{P}_1 \sim \mathcal{W}_p(m_1, \Sigma_1)$ amounts to

$$\mathrm{KL}(\mathbb{P}_2 \,\|\, \mathbb{P}_1) = -\frac{m_1}{2} \log \det(\Sigma_1^{-1} \Sigma_2) + \frac{m_2}{2}\big( \mathrm{Tr}[\Sigma_1^{-1} \Sigma_2] - p \big) + \log \frac{\Gamma_p(\frac{m_1}{2})}{\Gamma_p(\frac{m_2}{2})} + \frac{m_2 - m_1}{2} \psi_p(\frac{m_2}{2}).$$

For any generic prior distribution $\mathbb{P}_{t-1} \sim \mathcal{W}_p(m_{t-1}, \Sigma_{t-1})$, the above closed-form expression of the KL divergence helps us define the objective function

$$
\begin{aligned}
L(m, \Sigma) \quad &= -\frac{m_{t-1}}{2} \log \det(\Sigma_{t-1}^{-1} \Sigma) + \frac{m}{2}\big( \mathrm{Tr}[\Sigma_{t-1}^{-1} \Sigma] - p \big) + \log \frac{\Gamma_p(\frac{m_{t-1}}{2})}{\Gamma_p(\frac{m}{2})} \\
&\quad + \frac{m - m_{t-1}}{2} \psi_p(\frac{m}{2}) - \tau \sum_{i \in \mathcal{I}_t^- \cup \mathcal{I}_t^+} R_{ti} F_{ti}(m, \Sigma),
\end{aligned}
\tag{4}
$$

where $F_{ti}(m, \Sigma) = \log \mathbb{P}(c_A(q_t, d_i) \leq 1)$. Because $A$ follows a Wishart distribution under $\mathbb{P} \sim \mathcal{W}_p(m, \Sigma)$, we have $(q_t - d_i)^\top A (q_t - d_i) \sim \sigma_{ti}(\Sigma)^2 \chi_m^2$ by the property of Wishart distribution (Rao, 2009, Section 8b.2), where $\chi_m^2$ is a chi-squared distribution with $m$ degrees of freedom and

$$\sigma_{ti}(\Sigma)^2 = (q_t - d_i)^\top \Sigma (q_t - d_i).$$

Thus one can evaluate $F_{ti}(m, \Sigma)$ using the closed form expression:

$$F_{ti}(m, \Sigma) = \log \mathrm{cdf}_m(1/\sigma_{ti}(\Sigma)^2),$$

where $\mathrm{cdf}_m$ is the cumulative distribution function of $\chi_m^2$. Consequently, problem (3) now becomes

$$\min \big\{ L(m, \Sigma) \,:\, m \in \mathbb{N}_+, \, \Sigma \in \mathbb{S}_{++}^p, \, p \leq m \leq m_{t-1} \big\},
\tag{5}$$

which is an optimization problem over the parameters of the Wishart posterior. Due to the integer structure of the parameter $m$, we can solve sequentially for each $m$ between $p$ and $m_{t-1}$. For a fixed value of $m$, we need to solve over the matrix $\Sigma$, and the reduced optimization problem has the form

$$\min_{\Sigma \in \mathbb{S}_{++}^p} \big\{ \ell(\Sigma) \triangleq -m_{t-1} \log \det(\Sigma) + m \, \mathrm{Tr}[\Sigma_{t-1}^{-1} \Sigma] - \tau \sum_{i \in \mathcal{I}_t^- \cup \mathcal{I}_t^+} R_{ti} F_{tim}(\Sigma) \big\},
\tag{6}$$

where we use $F_{tim}(\Sigma) = F_{ti}(m, \Sigma)$ to highlight that $m$ is fixed and the optimization variable in (6) is only $\Sigma$. Next, we analyze the differentiability of $U_{im}$. Using the chain rule, we find

$$\nabla F_{tim}(\Sigma) = -\frac{1}{\mathrm{cdf}_m(1/\sigma_{ti}(\Sigma)^2)} \mathrm{pdf}_m\Big(\frac{1}{\sigma_{ti}(\Sigma)^2}\Big) \frac{(d_i - q_t)(d_i - q_t)^\top}{\sigma_{ti}(\Sigma)^4},$$

where $\mathrm{pdf}_m$ is the probability density function of $\chi_m^2$. The gradient information of $\ell$, the objective function of problem (6), evaluates to

$$\nabla \ell(\Sigma) = -m_{t-1} \Sigma^{-1} + m \Sigma_{t-1}^{-1} - \tau \sum_{i \in \mathcal{I}_t^- \cup \mathcal{I}_t^+} R_i \nabla F_{tim}(\Sigma).$$

The complete strategy to obtain the posterior distribution by solving (3) is described as follows:

- For any integer value $m = p, \ldots, m_{t-1}$: solve problem (6) using a line-search projected gradient descent algorithm to obtain $\Sigma^\star(m)$. Note that the dependence of the optimal solution on the fixed value of $m$ is now made explicit. A specification of the line-search algorithm to solve (6) is provided in the appendix.

- Identify $(m^\star, \Sigma^\star(m^\star))$ that minimizes the objective function $L$, where $L$ is defined in (4). The posterior distribution $\mathbb{P}_t$ is set to $\mathcal{W}_p(m^\star, \Sigma^\star(m^\star))$.

## 4.2 DOCUMENT RETRIEVAL USING MUTUAL INFORMATION

The second component of Phase 2 is a mutual information-based retrieval mechanism upon receiving a query $q_t$ from the user. There remains two criteria: (i) relevance, and (ii) value of information about the user's preference $A_0$. For relevance, we impose an $n$ nearest document requirement, similar to Phase 1. For the value of information: given a generic incumbent belief $\mathbb{P}_{t-1} \sim \mathcal{W}_p(m_{t-1}, \Sigma_{t-1})$ about the user's weight matrix $A_0$, our system will search for the document that can give us the most significant amount of information about $A_0$. Consider a document with embedding $d_i$ and its associated random variable $\tilde{R}_{ti} = R(q_t, d_i, A)$ representing whether the user likes or dislikes the item, whereas the randomness in $\tilde{R}_{ti}$ is induced by the prior distribution $\mathbb{P}_{t-1}$ on $A$. The mutual information between $A$ and $\tilde{R}_{ti}$ can be written as

$$\mathrm{MI}(A, \tilde{R}_{ti}) = \mathrm{H}(\tilde{R}_{ti}) - \mathrm{H}(\tilde{R}_{ti}|A) = \mathrm{H}(\tilde{R}_{ti}),$$

where the conditional entropy $\mathrm{H}(\tilde{R}_{ti}|A)$ is zero because given $A$, the feedback $\tilde{R}_{ti}$ becomes a deterministic constant. Note that by definition of the preference mechanism in Section 2.1, $R_{ti}$ is a Bernoulli random variable with probability $\gamma_{ti}^+ = \mathbb{P}_{t-1}(c_A(q_t, d_i) \leq 1)$. Because our belief about $A$ is represented by a Wishart distribution, the quantity $c_A(q_t, d_i) = (q_t - d_i)^\top A(q_t - d_i)$ is a scaled chi-square distribution. More specifically, we have $c_A(q_t, d_i) \sim \sigma_{ti}^2 \chi_{m_{t-1}}^2$, where $\chi_m^2$ is a chi-squared distribution with degrees of freedom $m_{t-1}$, and $\sigma_{ti}^2 = (q_t - d_i)^\top \Sigma_{t-1}(q_t - d_i)$. Hence, $\gamma_{ti}^+ = \mathrm{cdf}(1/\sigma_{ti}^2)$, where cdf is the cumulative distribution function of $\chi_m^2$. The entropy $\mathrm{H}(\tilde{R}_{ti})$ is

$$\mathrm{H}(\tilde{R}_{ti}) = -\gamma_{ti}^+ \log_2(\gamma_{ti}^+) - (1 - \gamma_{ti}^+) \log_2(1 - \gamma_{ti}^+),$$

and it is maximized when $\gamma_{ti}^+ = 0.5$. A reasonable approach to acquiring information is to retrieve documents with maximal mutual information value. Consequently, we propose finding $C_2$ documents that belong to the $n$ nearest document to the query $q_t$ and with the highest entropy value $\mathrm{H}(\tilde{R}_{ti})$. The system then processes these documents and shows the generated text sto the user, then, the user can provide feedback ($+1$ or $-1$) on these $C_2$ texts. The preference information can be fed into the posterior update in Section 4.1 to refine our belief.

## 5 PHASE 3: RETRIEVAL AND ANSWERING MODULE

After Phase 1 and Phase 2, we have collected sufficient information to learn about the user's preference. In Phase 3, we put less emphasis on refining our belief about $A_0$, but we focus on providing good quality answers to the user's query. Given the final posterior $\mathbb{P}_f \sim \mathcal{W}_p(m_f, \Sigma_f)$ and a user's query $q_t$, we can calculate our belief probability that the user favors document $d_i$ by $\mathbb{P}_f(c_A(q_t, d_i) \leq 1)$. This probability amounts to $\mathrm{cdf}(1/\sigma_{ti}^2)$, where cdf is the cumulative distribution function of $\chi_{m_f}^2$ and $\sigma_{ti}^2 = (q_t - d_i)^\top \Sigma_f(q_t - d_i)$. Here, $\sigma_{ti}^2$ depends on the matrix $\Sigma_f$ representing the preference deviation away from the nominal preference. Thus, the rank list can be obtained by simply sorting the nearest documents to $q_t$ by the decreasing values of $\mathrm{cdf}(1/\sigma_{ti}^2)$ and picking the top items in the sorted list. Because cdf is a monotonically increasing function, we can equivalently sort in terms of the $1/\sigma_{ti}^2$ values to reduce the computational burden.

**Answering model**: We consider formulating responses to users based on retrieving the top-$k$ documents as an abstractive summarization task. Consequently, we employ a pre-trained language model that incorporates an autoregressive decoder for the purpose of generating summarizations. These generated summarizations subsequently serve as responses to the user's queries.

## 6 EXPERIMENTS

### 6.1 DATA COLLECTION

In order to assess the effectiveness of our methodology and replicate the scenario outlined in previous sections, we aim to curate a dataset featuring numerous iterations of the same information presented through varying expressions and writing styles. With this objective in mind, we leverage ChatGPT Turbo 3.5 to generate a dataset, simulating a scenario in which multiple users interact with a chatbot system, requesting natural language explanations for various concepts. We have generated

three distinct datasets, each centered around user queries related to concepts from three distinct domains: politics, law, and technology. In our dataset, each user is designed to embody a unique facet of humanity, such as their profession or educational background. We have observed that one's profession most effectively captures this distinctive trait. Please refer to the Appendix for details about the prompt we use to synthesize our dataset.

Our dataset comprises three components: 1) User Types: user's professions that mirror the educational backgrounds of our users. 2) Concepts: these encompass various terms, including technological, political, and legal terminology, which we aim to elucidate for our users. 3) Documents: these documents serve the purpose of clarifying concepts for users, with content specifically customized to match their professions and educational backgrounds. We synthesize three datasets, each dataset comprises 50 distinct concepts, encompassing 100 diverse user types, and is composed of a corpus of 5000 documents.

## 6.2 EXPERIMENTAL SETUP AND EVALUATION METRICS

We design the experiments to answer the following research question: Can our Bayesian learning framework effectively capture user preferences for different document writing styles, even with a limited number of feedbacks?

**Baselines**: We evaluate our method against three commonly used baseline retrievers in the RAG literature, including sparse retrievers, BM25+ Lv & Zhai (2011), and dense retrievers, Maximizing Inner-Product Search (MIPS) Lewis et al. (2020) and Euclidean $l_2$ distance.

**Evaluation:** we employ the leave-one-out cross-validation method to assess the effectiveness of our approach. For each user type, we systematically exclude all documents associated with that user type from the database, treating them as reference documents. Subsequently, we identify the nearest $C$ documents as the designated ground truth documents for evaluation purposes. Our evaluation encompasses two key aspects of our method. Firstly, we utilize the retrieval module to extract the top-$k$ documents that align most closely with the user. We conduct evaluations at $k = 1$ and $k = 3$. Secondly, we assess the generative module by inputting the top-3 documents into an abstractive summarization model to generate an explanation. This explanation should encompass the concept's definition and user-specific analogies.

**Model architecture:** For query and document embedding, we use MiniLM Wang et al. (2020)[1]. For summarization model, we use bart-large-cnn Lewis et al. (2019) [2]. Details about the hyperparameters lie in the Appendix.

We choose the following values for the technical parameters: the dimension of the text embedding space is set to $p = 128$. We update the Bayesian posterior with parameter $\tau = 1$ in Section 4.1. The main paper presents the experiments with $N_c = 3$, while experiments with $N_c = 5$ are included in the Appendix.

We employ two categories of metrics to measure the retrieval and text generation quality.

**Retrieval Metrics.** We utilize Precision@K and Hit Rate (HR@K), both are common metrics in the domain of information retrieval.

**Text Generation Metrics.** We utilize a combination of Rouge scores (Lin, 2004), the $l_2$ distance, and a customized adaptation of the recent G-Eval (Liu et al., 2023) to gauge the degree of relatedness. We also measure an additional G-Eval metric because Rouge scores may not have a high correlation with human preference in evaluating tasks of a creative and diverse nature. Please refer to the Appendix for further details on each performance metric.

## 6.3 EXPERIMENTAL RESULTS

### 6.3.1 RETRIEVAL QUALITY

Table 1 presents the outcomes of our retrieval performance evaluation, focusing on Precision and Hit Rate metrics. Notably, our algorithm outperforms all baseline methods by a substantial margin. This

---

[1]Accessible at `https://huggingface.co/sentence-transformers/all-MiniLM-L6-v2`
[2]Accessible at `https://huggingface.co/facebook/bart-large-cnn`

improvement margin widens as we grant our method more opportunities to solicit user feedback. These findings strongly indicate that our perturbed Mahalanobis distance effectively captures user preferences, despite the limited availability of user's feedback for our method.

| Dataset | Methods | Precision@1 ↑ | Precision@3 ↑ | HR@3 ↑ |
|---|---|---|---|---|
| Politics | BM25+ | 0.1109 | 0.103 | 0.2622 |
| | Euclidean | 0.3141 | 0.3347 | 0.6367 |
| | MIPS | 0.2585 | 0.2564 | 0.5319 |
| | **Ours** | **0.3704** | **0.3751** | **0.6849** |
| Law | BM25+ | 0.0749 | 0.0947 | 0.2493 |
| | Euclidean | 0.3168 | 0.2926 | 0.5849 |
| | MIPS | 0.1828 | 0.1849 | 0.4221 |
| | **Ours** | **0.3503** | **0.3199** | **0.6176** |
| Technology | BM25+ | 0.092 | 0.0672 | 0.194 |
| | Euclidean | 0.454 | 0.3574 | 0.6642 |
| | MIPS | 0.204 | 0.2235 | 0.4975 |
| | **Ours** | **0.4621** | **0.3715** | **0.6803** |

Table 1: Quality of retrieval module with different retrieval methods. The best performance for any metric is highlighted in bold. ($C_1 = C_2 = 3$, $N_c = 3$)

### 6.3.2 GENERATED ANSWER QUALITY

Table 2 presents the empirical results concerning the text quality of the generated summaries. Our approach stands out by achieving the lowest $l_2$ distance and the highest Rouge scores when compared to BM25+, Euclidean, and MIPS. This performance indicates that our method generates responses that closely align with the semantic content of the user's query. Additionally, our method attains the highest averaged G-Eval scores, suggesting that our generated answers are more likely to align with the user's preferences.

| Dataset | Methods | $l_2$ ↓ | Rouge1 ↑ | Rouge2 ↑ | RougeL ↑ | RougeLSum ↑ | G-Eval ↑ |
|---|---|---|---|---|---|---|---|
| Political | BM25+ | 0.3232 | 0.3452 | 0.0923 | 0.1933 | 0.2570 | 2.8224 |
| | Euclidean | 0.257 | 0.3669 | 0.1077 | 0.2044 | 0.2711 | 3.1980 |
| | MIPS | 0.257 | 0.3671 | 0.1085 | 0.2060 | 0.2726 | 3.2486 |
| | **Ours** | **0.2513** | **0.3701** | **0.1109** | **0.2072** | **0.2734** | **3.2738** |
| Legal | BM25+ | 0.3496 | 0.3736 | 0.1098 | 0.2069 | 0.2728 | 2.7797 |
| | Euclidean | 0.267 | 0.3851 | 0.1196 | 0.2163 | 0.2828 | 3.0851 |
| | MIPS | 0.2726 | 0.3818 | 0.1166 | 0.2148 | 0.2818 | 2.9794 |
| | **Ours** | **0.2659** | **0.3868** | **0.1210** | **0.2173** | **0.2842** | **3.1189** |
| Technology | BM25+ | 0.3812 | 0.3422 | 0.0955 | 0.1941 | 0.2567 | 3.0283 |
| | Euclidean | 0.2886 | 0.3553 | 0.1050 | 0.2016 | 0.2687 | 3.5776 |
| | MIPS | 0.2816 | 0.3523 | 0.1037 | 0.2004 | 0.2659 | 3.4550 |
| | **Ours** | **0.2807** | **0.3654** | **0.1130** | **0.2024** | **0.2702** | **3.5940** |

Table 2: Quality of generated summary with different retrieval methods. The best performance for each metric is highlighted in bold. ($C_1 = C_2 = 3$, $N_c = 3$)

## 7 CONCLUSION

In this work, we propose a hybrid of Bayesian Learning and Retrieval Augmented Generation framework to capture the user's preference in the context of a retrieval-based text generation. We propose a preference learning framework that constructs of three elicitation phases. In Phase 1, we use a determinantal point process to sample $K$ diverse documents and get feedback from the user. In Phase 2, we sequentially acquire additional user's feedback, and employ a Bayesian framework to update the belief on user-specific preference metric. In the final phase, we prioritize delivering accurate and valuable responses to the user's query. Numerical experiments show that our method outperforms the baselines in terms of retrieval and text generation quality across multiple metrics.

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

## A  ADDITIONAL NUMERICAL RESULTS

| Dataset | Methods | Precision@1 ↑ | Precision@3 ↑ | HR@3 ↑ |
|---------|---------|---------------|---------------|--------|
| Politics | BM25+ | 0.1105 | 0.0956 | 0.2458 |
| | Euclidean | 0.3003 | 0.3193 | 0.6132 |
| | MIPS | 0.2493 | 0.2472 | 0.5198 |
| | **Ours** | **0.459** | **0.4297** | **0.7735** |
| Law | BM25+ | 0.0771 | 0.0929 | 0.2479 |
| | Euclidean | 0.3133 | 0.2848 | 0.5753 |
| | MIPS | 0.1828 | 0.1849 | 0.4221 |
| | **Ours** | **0.4777** | **0.4312** | **0.7492** |
| Technology | BM25+ | 0.088 | 0.063 | 0.1785 |
| | Euclidean | 0.413 | 0.3262 | 0.618 |
| | MIPS | 0.2065 | 0.2132 | 0.4745 |
| | **Ours** | **0.5475** | **0.4942** | **0.7995** |

Table 3: Quality of retrieval module with different retrieval methods, measured using different metrics. Bold indicates best performance. ($C_1 = C_2 = 3$, $N_c = 5$).

| Dataset | Methods | $l_2$ ↓ | Rouge1 ↑ | Rouge2 ↑ | RougeL ↑ | RougeLSum ↑ | G-Eval ↑ |
|---------|---------|---------|----------|----------|----------|-------------|----------|
| Political | BM25+ | 0.3288 | 0.3460 | 0.0928 | 0.1932 | 0.2575 | 3.0066 |
| | Euclidean | 0.2649 | 0.3676 | 0.1072 | 0.2039 | 0.2710 | 3.3926 |
| | MIPS | 0.2655 | 0.3681 | 0.1073 | 0.2050 | 0.2721 | 3.4273 |
| | **Ours** | **0.2466** | **0.3745** | **0.1112** | **0.2072** | **0.2760** | **3.4777** |
| Legal | BM25+ | 0.3533 | 0.3414 | 0.0946 | 0.1923 | 0.2567 | 2.9673 |
| | Euclidean | 0.2735 | 0.3537 | 0.1031 | 0.1996 | 0.267 | 3.2414 |
| | MIPS | 0.2782 | 0.3510 | 0.1036 | 0.1993 | 0.2653 | 3.1562 |
| | **Ours** | **0.2563** | **0.3694** | **0.1161** | **0.2100** | **0.2797** | **3.3490** |
| Technology | BM25+ | 0.3895 | 0.3414 | 0.0946 | 0.1923 | 0.2567 | 2.9952 |
| | Euclidean | 0.3002 | 0.3537 | 0.1031 | 0.1996 | 0.267 | 3.5822 |
| | MIPS | 0.2946 | 0.351 | 0.1036 | 0.1993 | 0.2653 | 3.4873 |
| | **Ours** | **0.2658** | **0.3694** | **0.1161** | **0.2100** | **0.2797** | **3.655** |

Table 4: Quality of generated summary with different retrieval methods. The best performance for any metric is highlighted in bold. ($C_1 = C_2 = 3$, $N_c = 5$).

## B  PERFORMANCE METRICS DETAILS

Below are details of the performance metrics we use throughout our evaluation.

- $l_2$: We use the Euclidean distance on the embedding space to measure the semantic similarity of the generated answer and the reference document.
- Rouge: is a set of metrics designed for summarization tasks. There are four variants of Rouge, which operates at different levels of document analysis: ROUGE-1 (utilizing unigram-based scoring), ROUGE-2 (employing bigram-based scoring), ROUGE-L (based on the Longest Common Subsequence), and ROUGE-Lsum (focusing on summary-level scoring).
- G-Eval: Following the recent work of G-Eval Liu et al. (2023), we create a variant of G-Eval using GPT-3.5-turbo and a customized prompt.

Below we present the prompt to evaluate the summarization quality using GPT-3.5-turbo, following Liu et al. (2023).

```
Imagine you are a {{user_type}} tasked with evaluating an explanation
provided for a query by another {{user_type}}. Your evaluation should
focus on one key metric: Relatedness (rated on a scale of 1-5). This
metric assesses how closely the analogy used in the explanation aligns
```

```
with the profession of {{user_type}}.

A score of 1 (far from relevant) indicates that the explanation is
generic and unrelated to {{user_type}}'s profession.

A score of 2 (somewhat relevant) suggests that the explanation contains
some relevance to {{user_type}}'s profession but lacks depth.

A score of 3 (moderately relevant) signifies that the explanation has a
reasonable connection to {{user_type}}'s profession.

A score of 4 (highly relevant) indicates that the explanation closely
aligns with {{user_type}}'s profession, making it easily understandable.
A score of 5 (perfectly relevant) means that
the explanation is exceptionally well-suited to {{user_type}}'s profession,
providing a clear and comprehensive understanding.
To perform this evaluation, follow these steps:

Thoroughly review the Query and the Explanation.
Rate the response on a scale of 1-5 for relatedness
based on the aforementioned criteria.
Offer a concise explanation for your rating,
referencing specific elements of the explanation and the query.
Query: {{Query}}
Explanation: {{Explanation}}
Evaluation Form (scores ONLY):
Relatedness:
```

### B.1 DATASET CREATION

We use GPT-3.5-turbo to populate our dataset with the following prompt:

```
You will be presented with a {concept_type} concept.
Your task is to elucidate this concept to a
specific {user_type}, tailoring your language,
examples, and context to align with the professional
background of the {user_type}.
Your explanation should be concise and not exceed 200 words.
```

The list of concept types and user types is included in the supplementary material.

## C ALGORITHMS

We use this appendix to describe the heuristics used for diverse sampling and the algorithm for the posterior update.

### C.1 HEURISTIC ALGORITHM FOR MAXIMUM-A-POSTERIORI SOLUTION OF DPP

A popular algorithm to find the approximate MAP estimator in problem (2) is the greedy heuristics. It starts by initializing the set $J$ to an empty set. It then proceeds at each iteration by finding a greedy index $j$ to add to the incumbent set of documents $J$, with $j$ maximizes the incremental log-determinant value:

$$j = \arg\max_{i:i\notin J} \ \log\det(L_{J\cup\{i\}}) - \log\det(L_J),$$

The algorithm then adds $j$ to the set of documents until the cardinality constraint is met, that is, when $|J| = C_1$. This greedy algorithm costs $\mathcal{O}(C_1^2 n)$, and an implementation is provided in Chen et al. (2018).

**Local neighborhood search.** The terminal greedy heuristic solution may get stuck at a local minimum. To improve the solution quality, we can employ a simple 2-neighborhood local search that switches one element from the incumbent set with one element from the complementary set if this switch leads to a better objective value. The local neighborhood search is terminated if no improving switch can be found.

## C.2 OPTIMIZATION ALGORITHM FOR POSTERIOR UPDATE

Here we present a simple line-search gradient descent algorithm to solve the nonlinear semidefinite problem (6). The algorithm uses Armijo's rule to find the step size that guarantees the feasibility of the solution and sufficient descent.

---

**Algorithm 1:** Line-search Gradient Descent for Problem (6)

---

**Input:** Prior matrix $\Sigma_{t-1} \in \mathbb{S}^p_{++}$ and degree of freedom $m_{t-1}$, incumbent value $m$
**Parameter:** Number of iterations: $K \in \mathbb{N}_+$, Line search parameter $\gamma = 0.5$, $\beta = 10^{-4}$
**Initialization:** Set $S_{(0)} \leftarrow \Sigma_{t-1}$
   **for** $k = 0, \dots, K - 1$ **do**
      Compute gradient: $g \leftarrow -m_{t-1} S_{(k)}^{-1} + m \Sigma_{t-1}^{-1} - \tau \sum_{i \in \mathcal{I}_t^- \cup \mathcal{I}_t^+} R_{ti} \nabla F_{tim}(S_{(k)})$
      Find the minimum non-negative integer $\eta$ such that: (i) Feasibility: $S_{(k)} - \gamma^\eta g \in \mathbb{S}^p_{++}$, (ii)
      Descent guarantee:
$$\ell(S_{(k)} - \gamma^\eta g) \leq \ell(S_{(k)}) - \beta \gamma^\eta \|g\|_F^2$$
   **end for**
**Output:** $S_{(K)}$

---

## C.3 IMPLEMENTATION DETAILS:

Our implementation will be public at: `https://anonymous.4open.science/r/BayesianRAG-FDAF`

Hyperparameters are summarized in Table 5.

| Text Embedding | Dimension $p$ | 128 |
|---|---|---|
| Summarization | token min | 70 |
| | token max | 200 |
| Bayesian Learning | $\tau$ | 1 |
| | $C_1$ | 3 |
| | $C_2$ | 3 |
| | $N_c$ | 3 |

Table 5: Hyper-parameter used throughout our experiments. $C_1$ and $C_2$ are the number of elucidating concepts used in Phase 1 and Phase 2, respectively. $N_c$ is the number of user elicitation per concept in Phase 1 and Phase 2

