# OpenReview forum: "Personalized Language Generation via Bayesian Metric Augmented Retrieval"
_ICLR.cc/2024/Conference — Submitted to ICLR 2024_

### Official Review · Reviewer_NT9n · 2023-10-29

**Soundness:** 2 fair
**Presentation:** 2 fair
**Contribution:** 2 fair
**Rating:** 5
**Confidence:** 5

**Summary:**

This paper proposes a hybrid of Bayesian Learning and Retrieval Augmented Generation (RAG) framework to capture the characteristics of each user using specific perturbations of the local metric of the embedding space.
Authors aim to construct a comprehensive three-phase solution package for RAG-based conversational systems.
Its primary aim is to enable the retrieval of documents that closely align with human preferences,
ultimately enhancing the quality of text generation.
They demonstrate that their approach captures the user’s preference, and consequentially improve the retrieval and text generation quality under a llimited number of queries.

**Strengths:**

**Bayesian Learning and Retrieval Augmented Generation (RAG) framework is simple and reasonable**
* Proposed solution system is simple and reasonable for adapting the retrieval mechanism to the the user’s preferences.
* To capture the user’s preference, encompass factors such as their educational background and professions,and consequentially improve the retrieval and text generation quality, authors model these preference through the metric drift of the embedding space, which governs the ranking of the documents for each input query.
* Learning a Bayesian posterior estimate of the metric as the addtional layer is designed to empirical improvement in the retrieval quality and in the performance of the generated text on multiple concept explanation tasks.

**Weaknesses:**

**The authors have considered and designed their system well, but their effectiveness, advantages and claims are weak.**

- While authors aim to capture the characteristics of each user,  they show that experimental results are search and textual quantitative evaluations, which do not lead to determine the level of achievement for this objective.

- The performance increase in Table 1 and 2 appears marginal and weak to support the claim.

- Since there is no ablation analysis, it is not possible to determine the effectiveness of the authors' proposed elements such as perturbed Mahalanobis metric, and Bayesian posterior estimate.

- This solution is a kind of RAG frameworks, but there is no comparison with other RAG based framework, so its superiority cannot be determined.

**Questions:**

* Did you not conduct a qualitative evaluation?
* Did you perform statistical tests on Table 1 and 2? If soIf so, please provide details.
* Please see Weaknesses.

---

> ### Author Response · Authors · 2023-11-22
> **Response to Reviewer NT9n**
>
> Dear reviewer,
>
> We appreciate that you review our paper and provide insightful comments. We would like to address your concerns as follows:
>
> > Did you perform statistical tests on Table 1 and 2? If so, please provide details.
>
> We propose to look at the paired difference of the performance metrics: for each subject, we compute our method's HR@3, Rouge1 and the competing method’s HR@3, Rouge1. We propose to test the hypotheses:
>
> Null hypothesis: Ours HR@3 and Rouge1 score equal competing method’s HR@3 and Rouge1
>
> Alternative hypothesis: Ours HR@3 and Rouge1 are greater than competing method’s HR@3 and Rouge1.
>
> In order to test the above hypothesis, we use a one-sided Wilcoxon signed-rank test to compare the paired performance metric values. The p-value of the test between our method and the baselines is reported in the following table.
>
> Suppose we choose the significant level at 0.05. The table indicates that our method significantly outperforms baselines across all datasets.
>
> | | Datasets  | Politics               |   Law             | Technology |
> |---------- |----------|--------------------|---------------------|---------------------|
> |HR@3 |Ours - Euclidean | 1e-15 | 8e-16  | 8e-14 |
> | |Ours - BM25 | 2e-18 | 2e-17 | 2e-18 |
> | |Ours - MIPS | 7e-18  | 2e-17 | 2e-16 |
> | |Ours - Logistic Regression | 7e-16  |5e-17 | 4e-16 |
> |Rouge1 |Ours - Euclidean | 1e-12 | 1e-14 | 5e-15 |
> | |Ours - BM25 | 1e-18 | 6e-17 | 2e-18 |
> | |Ours - MIPS | 3e-10  | 4e-16 | 1e-14 |
> | |Ours - Logistic Regression | 4e-17  | 7e-18 | 3e-17 |

---

> > ### Comment · Reviewer_NT9n · 2023-11-23
> > **Reply to authors**
> >
> > Dear,
> >
> > Thank you for your response.
> > While this table shows statistical differences,
> > but please clarify more if there are qualitative differences or error analysis.

---

> ### Author Response · Authors · 2023-11-23
> **Response to Reviewer NT9n**
>
> Dear Reviewer, thank you for your feedback on our submission. Regarding the missing qualitative evaluation, we would like to provide an illustrative example from the **Law dataset**. Consider the query 'Explain Criminal Law to me', posed by a user whose profession is a `Dentist`. Our system knows nothing of this user but a few of his liked or disliked documents (a total of 18 documents in our experiment). Our system uniquely produces documents with analogies pertinent to medical or health-related fields, such as `patients` or `treatment or therapy`. This illustrates how our method tailors content to user-specific contexts, even with limited information about the user.
>
> **Our method**:
> ```
> Criminal law refers to the body of laws that define and regulate behaviors that are considered to be crimes. You may encounter the concept of criminal law in your practice when dealing with **patients** who have engaged in illegal activities. Your expertise can be crucial in providing insights into their mental capacity, potential mitigating factors, or the need for **treatment or therapy**.
> ```
>
> **Euclidean**:
> ```
> Criminal law is a branch of law that deals with offenses committed against society as a whole, rather than individual disputes. This is similar to how physics principles, like the laws of motion, help maintain order and predict the movement of objects in the universe.
> ```
>
> **BM25**:
> ```
> Criminal Law is like a set of guidelines that defines what actions are considered against the law and can result in punishment. Just like there are certain rules and principles you follow while creating your beautiful artwork, there are rules in society that everyone must abide by. You may find Criminal Law relevant to your work in developing AI systems that interact with society.
> ```
> **MIPS**:
> ```
> Criminal law is a branch of law that deals with offenses committed against society as a whole, rather than individual disputes. By incorporating the principles of criminal law into your storytelling, you can create suspenseful plots, explore moral dilemmas, and delve into the psychological motivations behind criminal behavior.
> ```

---

> > ### Comment · Reviewer_NT9n · 2023-11-23
> > **Reply to authors**
> >
> > Thank you for showing example.
> >
> > The example of the LAW dataset is interesting,
> > but since only a limited number of readers are capable of making this qualitative judgment,
> > a more general example would be better.

---

> ### Author Response · Authors · 2023-11-23
> **Response to Reviewer NT9n**
>
> Dear reviewer NT9n,
>
> Here we show another example in the **Politics Dataset**, whose user identity and concept are more widely known. In this example, we observe that only our method generates content that is aligned with the common objective of an `online content creator`, which is to `build trust and loyalty` or `appeal to your target audience`
>
> **True user identity**: Online Content Creator
>
> **Query**: "Explain women's reproductive rights"
>
> **Our method**:
> ```
> Women's reproductive rights include access to contraception, family planning, and maternity care. They also include the freedom to make choices about when to have children and how many. Some places have restrictions on access to reproductive healthcare, limiting women's choices and jeopardizing their well-being. By highlighting women's rights, you can **appeal to your target audience of women** who want to make decisions that align with their personal goals and aspirations. In summary, understanding and respecting women's reproductive rights can **build trust and loyalty**.
> ```
>
> **Euclidean**:
> ```
> Reproductive rights refer to a set of freedoms that women have regarding their reproductive health and choices. The lack of access to contraception and abortion services has historically limited women's ability to control their fertility and plan their families. By supporting women's reproductive rights, we promote equality and empower women to have control over their own lives. By studying the struggles and triumphs of the past, historians can provide valuable insights into the evolution of women’s reproductive rights.
> ```
>
> **BM25**:
> ```
> Women have the right to make decisions about their own reproductive health and rights. You can help by examining the impact of different levels of reproductive rights. For example, you can examine the effect of different types of birth control on the rate of unwanted pregnancies. You could also ask questions about the effectiveness of different countries' reproductive rights laws, such as those that allow women to choose not to give birth to their children in the first place.
> ```
> **MIPS**:
> ```
> Women's reproductive rights are crucial because they ensure that women have autonomy over their bodies. The lack of access to contraception and abortion services has historically limited women's ability to control their fertility and plan their families. By studying the struggles and triumphs of the past, historians can provide valuable insights into the evolution of women’s reproductive rights. By recognizing and respecting women's reproductive rights, we empower them to take control of their lives, pursue their agricultural endeavors, and contribute to the sustainable development of our communities.
> ```

---

### Official Review · Reviewer_Qd94 · 2023-11-01

**Soundness:** 2 fair
**Presentation:** 3 good
**Contribution:** 2 fair
**Rating:** 5
**Confidence:** 3

**Summary:**

The paper proposes an approach for learning a personalized ranking model followed by generation of personalized summaries using an off the shelf summarization model. The paper proposes a probabilistic model for learning user preferences where user preferences are modeled as a perturbed Mahalanobis metric. The paper proposes a 2 stage method for learning user preferences: in Phase 1 a detreminantal point process is used to retrieve diverse documents and gather user preferences for the documents given queries, in Phase 2 gathered feedback is used to make an update to the user preferences. In Phase 3 the learned user preferences are used for ranking documents for the query and generating summarizations of the ranked documents. The proposed approach is evaluated on 3 synthetic datasets generated with gpt-3.5-turbo with 50 concepts (queries), 100 user types (users), resulting in retrieval over 5000 documents.

**Strengths:**

- The paper proposes an interesting Bayesian framework for learning personalized ranking models.
- The experimental evaluations seem to indicate the value of the proposed framework.
- The paper is largely well-written.

**Weaknesses:**

- Simplistic synthetic data: The only data used for experiments are synthetic with the conceived form of user preference being somewhat simplistic (different user types seeking stylistically different concept definitions).
- Weak baselines: The baselines compared against are non-personalized and don't use user type information, one would expect to see poorer performance from such methods given the setup of the paper.
- Unclear experimental setup.
- The paper is ungrounded from prior work making it hard to see what aspects of the contribution are novel.

**Questions:**

- Am I understanding correctly that the 5000 large document collection contains the 50 concept definitions rephrased in 100 different ways?
- Am I understanding correctly that the proposed approach never uses the pre-trained summarization model in learning user preferences?
- Is the user feedback for Phase 2 obtained from labelled data? Eg if the document corresponds to the correct concept+user it is a positive document and negative if not?
- Please clarify in greater detail what data is used for training and evaluation. "For each user type, we systematically exclude all documents associated with that user type from the database, treating them as reference documents" -- its not clear what this means? Is this implying that no documents corresponding to the user type are available for training?
- Sec 6.1: What reference documents are used for summarization evaluation? Eg in computing metrics like rouge.
- The paper could benefit from inclusion of examples for concept, user type, positive document and negative document for concept and user type - perhaps in Fig 1.
- If the experimental setup contains as input a query + positive document for user type - It would be illustrative to have a simple personalized ranking baseline which computes a score based on a simple linear interpolation concept-document score and a concept-user_type score. Perhaps a user_type could be represented with the average of document embeddings for that user and the weight for the two components tuned on a dev set. Alternatively, please consider formulating an appropriate, simpler personalized ranking baseline and comparing to it in addition to the ones in the paper.
- The paper contains no context on related work from personalized search, it would be useful to make some connection to related prior work. Starting points may be: https://dl.acm.org/doi/10.1145/1076034.1076111, https://dl.acm.org/doi/10.1145/3357384.3357980, https://dl.acm.org/doi/10.1145/2124295.2124348. Also consider discussing differences to the task/approach of: https://arxiv.org/abs/2211.09260

---

> ### Author Response · Authors · 2023-11-22
> **Response to Reviewer Qd94**
>
> Dear reviewer,
>
> We appreciate that you review our paper and provide insightful comments. We would like to address your concerns as follows:
>
> > Am I understanding correctly that the 5000 large document collection contains the 50 concept definitions rephrased in 100 different ways?
>
> Yes, it is correct. As mentioned in Section 6.1, we synthesize 3 datasets (Political, Legal and Technology) with that setup.
>
> > Am I understanding correctly that the proposed approach never uses the pre-trained summarization model in learning user preferences?
>
> Yes, it is correct. However, we did use a pretrained summarization model to shorten documents for practical purposes.
>
> > Is the user feedback for Phase 2 obtained from labelled data? Eg if the document corresponds to the correct concept+user it is a positive document and negative if not?
>
> A user likes a document if that document lies in the top 5 documents that are closest to the user’s true document in terms of Euclidean distance; A user dislikes a document otherwise.
>
> > Please clarify in greater detail what data is used for training and evaluation. "For each user type, we systematically exclude all documents associated with that user type from the database, treating them as reference documents" -- its not clear what this means? Is this implying that no documents corresponding to the user type are available for training?
>
> Yes, the document corresponding to the user type will not be used for training but only used as a reference document to compute the top-5 documents that will be liked by the user.
>
> > Sec 6.1: What reference documents are used for summarization evaluation? Eg in computing metrics like rouge.
>
> The document corresponding to the user type will be used as a reference document to compute rouge scores for the generated answer.
>
> > If the experimental setup contains as input a query + positive document for user type - It would be illustrative to have a simple personalized ranking baseline that computes a score based on a simple linear interpolation concept-document score and a concept-user_type score. Perhaps a user_type could be represented with the average of document embeddings for that user and the weight for the two components tuned on a dev set. Alternatively, please consider formulating an appropriate, simpler personalized ranking baseline and comparing to it in addition to the ones in the paper.
>
> We consider fitting user preference feedback to a simple logistic regression binary classifier. Please refer to the table below for results on all three datasets.
>
> |  Dataset  | Methods               |  HR@3             | Precision@3 | Rouge1 | Rouge2|
> |----------|--------------------|---------------------|---------------------|---------------------|---------------------|
> | Politics| Logistic Regression | 0.59 | 0.3036  | 0.3472  |  0.0975 |
> | | Ours | **0.6849** | **0.3751**  |  **0.3701**  |  **0.1109** |
> | Law | Logistic Regression |  0.4809 | 0.2521  | 0.3640  |  0.0933 |
> | | Ours | **0.6176** | **0.3199** |  **0.3868**  |  **0.1210** |
> | Technology | Logistic Regression |   0.5485 | 0.2704  | 0.3455  |  0.0950 |
> | | Ours | **0.6803** | **0.3715**  |  **0.3654**  |  **0.1130** |
>
> > Also consider discussing differences to the task/approach of: https://arxiv.org/abs/2211.09260
>
> The work in Task-aware Retrieval with Instructions by Asai et al. (2022) adapts its search results using an additional search task instruction $t$ in natural language, along with the user's query $q$ (see Section 3, 'Task Formulation'). Our work differs from that of Asai et al. (2022) in that we rely on the users' ratings (like/dislike) on a selection of summarized documents, instead of using the task instruction $t$.

---

> > ### Comment · Reviewer_Qd94 · 2023-11-23
> > **Thank you for the clarifications.**
> >
> > Thank you for the clarifications. I continue to echo the concerns of reviewer u3r5 -- the synthetic data setup of the paper is a non-trivial weakness of the paper.

---

> > > ### Author Response · Authors · 2023-11-23
> > > **Response to Reviewer Qd94**
> > >
> > > Thank you for your feedback on our response. Regarding your concerns about the synthetic setup of our approach, we wish to underscore that our paper's primary objective is to generate text aligning with the preferences of users. Our experimental approach is supported by multiple studies indicating the similarity between text generated by ChatGPT and human-generated text:
> > >
> > > - Is GPT-3 Text Indistinguishable from Human Text? Scarecrow: A Framework for Scrutinizing Machine Text (Dou et al., ACL 2022)
> > > - All That’s ‘Human’ Is Not Gold: Evaluating Human Evaluation of Generated Text (Clark et al., ACL-IJCNLP 2021)
> > >
> > > Consequently, simulating human-like responses using ChatGPT in our experiments is a reasonable approach.

---

### Official Review · Reviewer_u3r5 · 2023-11-03

**Soundness:** 2 fair
**Presentation:** 2 fair
**Contribution:** 2 fair
**Rating:** 3
**Confidence:** 4

**Summary:**

This paper introduces a Bayesian approach to modeling user preferences for document retrieval to be used in RAG, outperforming a few baselines on a generated dataset.

**Strengths:**

The paper tackles a timely and interesting problem of user preferences in RAG. The bayesian approach is reasonable and thoroughly described. The multi-phase setup is logical. The paper is generally well-written.

**Weaknesses:**

I have two main issues with the manuscript: The overly complex modeling procedure and the synthetic dataset.

While I am a fan of Bayesian approaches, there is no appropriate ablation to validate the inherent complexity. Why not just fit a simple linear model to the user preferences?
Many of the design choices in the proposed model are not justified. Why are we modeling the user preference as a a Wishart probability distribution? Why do we need a determinantal point process? How would that compare to random sampling? Et cetera.

The other issue is the dataset: it's entirely synthetic. There is no evidence that the proposed model would be useful at reflecting real users preferences. If the evaluation had fake data and real users, that would be interesting and such a user study would be illuminating to the proposed work. If the evaluation had real data and fake users (as imagined by an LLM), that might be okay too as LLMs have shown signs that they can model generic user behavior. Ideally the proposed model would be used on real data with user. And as I have yet to see the combined use of synthetic data and synthetic users in a promising fashion that represents real users/data, I cannot recommend accept.

**Questions:**

What happens when you fit a simple linear model to users original preferences and update in an online fashion?

---

> ### Author Response · Authors · 2023-11-22
> **Response to Reviewer u3r5 [1/2]**
>
> Dear reviewer,
>
> We appreciate that you review our paper and provide insightful comments. We would like to address your concerns as follows:
>
> > What happens when you fit a simple linear model to users original preferences and update in an online fashion?
>
> As mentioned in Section 6.2, we only ask users to rate a total of 18 shortened documents (N_c = 3, C1 = 3, C2 = 3). This number of samples is much too low for fitting a simple logistic regression binary classifier (with two classes, `like` or `dislike`) and will lead to overfitting. We, anyways, have implemented the baseline that you proposed, in which we learn a logistic regression model to predict whether the user likes or dislikes each document. We conduct the comparison on the Politics, Law and Technology dataset. The table below presents the performance between two models.
> We can clearly observe that our proposed method dominates the simple logistic regression baseline.
>
> |  Dataset  | Methods               |  HR@3             | Precision@3 | Rouge1 | Rouge2|
> |----------|--------------------|---------------------|---------------------|---------------------|---------------------|
> | Politics| Logistic Regression | 0.59 | 0.3036  | 0.3472  |  0.0975 |
> | | Ours | **0.6849** | **0.3751**  |  **0.3701**  |  **0.1109** |
> | Law | Logistic Regression |  0.4809 | 0.2521  | 0.3640  |  0.0933 |
> | | Ours | **0.6176** | **0.3199** |  **0.3868**  |  **0.1210** |
> | Technology | Logistic Regression |   0.5485 | 0.2704  | 0.3455  |  0.0950 |
> | | Ours | **0.6803** | **0.3715**  |  **0.3654**  |  **0.1130** |
>
> > Why are we modeling the user preference as a Wishart probability distribution?
>
> Because a popular approach for document retrieval is the nearest neighbor approach. The standard, off-the-shelf implementation uses an Euclidean distance. Thus, a reasonable approach to cater the user preference in the retrieval procedure is to alternate the distance. We thus need to parametrize the distance so that we can perform the learning process. Mahalanobis distance is the most natural way to generalize the Euclidean distance, and a Mahalanobis distance is parametrized by a positive semidefinite matrix (this is the matrix $A$ in our paper). To invoke a Bayesian method, we need to impose a prior and a posterior on the parameter space, and the parameter space in our case is the set of all a positive semidefinite matrices. And the Wishart distribution is, again, the most natural distribution supported on the positive semidefinite matrix space. We hope that this line of argument clarifies your concern.
>
> > Why do we need a determinantal point process? How would that compare to random sampling?
>
> Detrimental Point Process is needed because when a user arrives, there is no user preference information that we can use to personalize the retrieval process. This is the cold-start setting, and diversifying the retrieval via DPP is a well-known and powerful technique in this cold-start situation. Many papers have demonstrated that DPP delivers better performance than random sampling, including:
> [1] Wang et al. (2020) Personalized Re-ranking for Improving Diversity in Live Recommender Systems, arXiv:2004.06390
> [2] Liu et al. (2022) Determinantal Point Process Likelihoods for Sequential Recommendation, arXiv:2204.11562
> [3] Wilhelm et al. (2018) Practical Diversified Recommendations on YouTube with Determinantal Point Processes.
> When we design our framework, we have already optimized the choice of the component. That is why we use a determinantal point process. We will add this argument to the revision.

---

> ### Author Response · Authors · 2023-11-22
> **Response to Reviewer u3r5 [2/2]**
>
> > The other issue is the dataset: it's entirely synthetic. There is no evidence that the proposed model would be useful at reflecting real users preferences.
>
> We do not fully understand the concern here.
>
> Our ultimate goal is to generate high-quality text (meaning that the generated text can be aligned better with the user’s preference). We do not have to reflect 100 percent correctly the user’s preferences in order to generate high-quality text. Our model is a *model*, it is an abstraction of the real world, it may be wrong, but it is useful.
>
> There are a plethora of examples that follow the same argument, let us describe a few:
>
> Discrete choice: The field of discrete choice builds abstract models that can help to predict the user’s choice behaviors (whether the person buys a certain product or not). For sure, it is *im*possible to reflect accurately the users preferences, nevertheless, discrete choice models can have reasonably good performance in predicting the user’s buy/not buy decision. Discrete choice models are also used a lot in marketing to personalized advertisement. Unfortunately, discrete choice model also require a large number of samples and/or contextual information to perform well, and does not fit into our setting.
> Portfolio allocation: It is also impossible to learn the risk appetite of the investors. Nevertheless, it does not stop us from building proxies (different utilities functions), that can help us optimize the portfolio allocation.
>
> Regarding the synthetic data concern: there are a growing literature on using GPT to generate datasets, for both training and testing. We list a few here:
>
> [ToxiGen: A Large-Scale Machine-Generated Dataset for Adversarial and Implicit Hate Speech Detection](https://aclanthology.org/2022.acl-long.234) (Hartvigsen et al., ACL 2022)
>
> [A Synthetic Data Generation Framework for Grounded Dialogues](https://aclanthology.org/2023.acl-long.608) (Bao et al., ACL 2023)
>
> [Self-Instruct: Aligning Language Models with Self-Generated Instructions](https://aclanthology.org/2023.acl-long.754) (Wang et al., ACL 2023)
>
> [Is GPT-3 Text Indistinguishable from Human Text? Scarecrow: A Framework for Scrutinizing Machine Text](https://aclanthology.org/2022.acl-long.501) (Dou et al., ACL 2022)

---

> > ### Comment · Reviewer_u3r5 · 2023-11-23
> > **I will keep my score**
> >
> > Thanks authors for the extra experiments. It is nice to see an improvement over a simple baseline.
> >
> > However, after seeing the other discussion I am inclined to keep my score. The synthetic dataset is a complete show stopper for me. While synthetic datasets are clearly useful in the NLP community, the combination of a brand new proposed method on a brand new synthetic dataset does not indicate to me generalality of either. I encourage the authors to evaluate both with real humans for a future version of the paper.

---

### Official Review · Reviewer_zTRj · 2023-11-05

[review text omitted: it was posted to a different submission]

---

> ### Comment · Reviewer_zTRj · 2023-11-22
> **Misalignment of review and manuscript**
>
> Accidentally the review appearing here is not for this paper but for an other paper assigned to me. The review fro this paper "Personalized Language Generation via Bayesian Metric Augmented Retrieval" is also linked to an other paper.

---

> > ### Author Response · Authors · 2023-11-22
> > **Thank you!**
> >
> > We appreciate your time and effort in reviewing for ICLR2024!

---

### Meta-Review · Area_Chair_4oFx · 2023-12-06

**Metareview:**

a)  The paper proposes an approach for learning a personalized ranking model followed by generation of personalized summaries using an off the shelf summarization model. The paper proposes a probabilistic model for learning user preferences where user preferences are modeled as a perturbed Mahalanobis metric.   The model is a hybrid of Bayesian Learning and Retrieval Augmented Generation (RAG).  Experiments.  They use determinantal point process for diversity and a Wishart for user preferences
b)  interesting combinations of probabilistic theories, well written
c)  poor selection of datasets left reviewers wondering, poor baselines, paper needs better empirical work.

**Justification For Why Not Higher Score:**

consensus on reject

**Justification For Why Not Lower Score:**

N/A

---

### Decision · Program_Chairs · 2024-01-16

Reject